# Role of Extracellular Mycobacteria in Blood-Retinal Barrier Invasion in a Zebrafish Model of Ocular TB

**DOI:** 10.3390/pathogens10030333

**Published:** 2021-03-13

**Authors:** Santhosh Kumar Damera, Ranjan Kumar Panigrahi, Sanchita Mitra, Soumyava Basu

**Affiliations:** 1Allhadini Mishra Zebrafish Facility, Brien Holden Eye Research Centre, L V Prasad Eye Institute (MTC Campus), Bhubaneswar 751024, India; santhosh23.biotech@gmail.com (S.K.D.); ranjanpanigrahi873@gmail.com (R.K.P.); 2Ocular Microbiology Service, L V Prasad Eye Institute (MTC Campus), Bhubaneswar 751024, India; sanchita.mitra@lvpei.org

**Keywords:** tuberculosis, ocular, dissemination, extracellular, macrophage, pathogenesis, zebrafish

## Abstract

Intraocular inflammation following mycobacterial dissemination to the eye is common in tuberculosis (TB)-endemic countries. However, the early host–pathogen interactions during ocular dissemination are unknown. In this study, we investigated the early events during mycobacterial invasion of the blood-retinal barriers (BRBs) with fluorescent-tagged *Mycobacterium marinum* (*Mm*), host macrophages, and retinal vasculature in a zebrafish model of ocular TB. We found that *Mm* invaded the vascular endothelium in either the extracellular or intracellular (inside phagocytes) state, typically 3–4 days post-injection (dpi). Extracellular *Mm* are phagocytosed in the retinal tissue and progress to form a compact granuloma around 6 dpi. Intracellular *Mm* crossing the BRBs are likely to be less virulent and either persist inside solitary macrophages (in most cases) or progress to loosely arranged granuloma (rarely). The early interactions between mycobacteria and host immune cells can thus determine the course of disease during mycobacterial dissemination to the eye.

## 1. Introduction

Ocular tuberculosis (TB) is a sight-threatening form of intraocular inflammation that is not only common in TB-endemic countries but is also being increasingly reported from non-endemic countries [1,2]. Despite its common occurrence, the pathogenesis of ocular TB has remained unclear [3]. Most of our understanding of the pathogenesis of this condition has emerged from histopathological studies of end-stage disease that required enucleation of the eyeball [4,5]. These studies have revealed the presence of granulomatous inflammation and acid-fast bacilli within ocular tissues, both of which unequivocally point to ocular dissemination of *Mycobacterium tuberculosis* (*Mtb*) in this disease. However, these studies do not provide any information on the early events during mycobacterial dissemination to the eye. The early events are critical, as they have the potential to either eliminate the infection completely or limit the establishment and/or expansion of granuloma, thereby influencing the clinical manifestation of the disease [6].

The zebrafish larva provides an opportunity to visualize the early host–pathogen interactions through fluorescent-tagged mycobacteria and host-immune cells [7,8]. It has several unique advantages, the most important being its transparency, which makes it amenable to live microscopy. The zebrafish and its natural pathogen, *Mycobacterium marinum* (*Mm*), have provided remarkable insights into possible mechanisms of host immune response and granuloma formation during human mycobacterial infection [9,10]. The zebrafish model has also been used to investigate mycobacterial infection in the central nervous system (CNS), especially the traversal of mycobacteria across the blood–brain barrier (BBB) [11,12,13]. It has also been used to study host–pathogen interactions in other CNS infections, such as *Cryptococcus neoformans* [14].

Recently, we submitted a report on a zebrafish model of ocular TB, in which we demonstrated mycobacterial localization and early granuloma formation in the vicinity of the blood-retinal barrier (BRB) following injection into the caudal veins of embryos [15]. We also demonstrated recruitment of peripheral blood monocytes into these early granulomas, suggesting breakdown of the BRB by the inflammatory focus. However, we did not investigate the events surrounding mycobacterial traversal across the BRB. In this study, we have further explored the mechanisms of mycobacterial invasion of the BRB and the influence of bacterial virulence on these events.

## 2. Results

### 2.1. Mm Causes High Rate of Ocular Infection Even with Low Inoculum of Systemic Infection

Our earlier study had demonstrated intraocular infection following caudal vein injections of 100 colony forming units (CFU) of *Mm* [12]. The dose was chosen in accordance with standard laboratory protocols. The inoculum used in the TB meningitis model ranged between 250 and 1040 CFU of *Mm* [15]. In the present study, we sought to demonstrate ocular dissemination with a lower inoculum of 25 CFU. The lower inoculum was expected to mimic human extra-pulmonary dissemination better than previous studies. We injected the larvae at 4 days post-fertilization (dpf) since the inner and outer BRB, as well as the BBB, have been found to be established at 3 dpf in earlier studies [16,17]. The 4-dpf time point was also used for caudal vein infection in the TB meningitis model [11,12]. We obtained an ocular infection rate of 60% (18 of 30 infected larvae, in two separate experiments) at 1 dpi (Figure 1A). The rate of ocular infection was defined as the percentage of *Mm*-injected fish showing presence of *Mm* in at least one of the eyes at 10× magnification under fluorescent microscopy. This remained between 60 and 64% until 6 dpi, even though there was a steady decrease in viable embryos due to the *Mm* infection.

Next, we evaluated the sequence of events that led to formation of early granuloma-like aggregates in the eye. Here, we used *Tg (mpeg1::BB)* larvae that express red fluorescent protein in macrophages. Our earlier study had demonstrated early granulomas comprising aggregates of infected and uninfected macrophages within the eyes of infected larvae [15]. The macrophages in these granulomas appeared to be derived from the circulating monocytes as well as from the resident macrophage population within the retina. In the current study, early granuloma formation was seen in six (33.3%) of the 18 eye-infected embryos. In eyes where the infections progressed to granuloma formation, we noted that between 1 and 3 dpi, *Mm* remained extracellular, either as single or multiple clumps (Figure 1B). Phagocytosis of *Mm* was first noted at 4 dpi in four eyes and at 5 dpi in another two eyes (Figure 1C). Additional macrophages were seen in the vicinity of infected macrophages at 5–6 dpi, probably as a result of chemotaxis (data not shown). Finally, at 6–7 dpi, the aggregation of macrophages into an early granuloma was noted (Figure 1D). The granulomas were typically compact, with tight arrangements of infected and uninfected macrophages.

In four eye-infected larvae, phagocytosis was first seen at 3 dpi, and *Mm* continued to remain inside solitary macrophages without progression to granuloma formation until 6 dpi (Figure 2A–D). In the remaining eight larvae, *Mm* were not seen beyond 3 dpi in the eyes. These *Mm* could have moved out of ocular circulation (as noted in our earlier study), or may have been removed by circulating phagocytes.

The dichotomy noted in the course of events between extracellular *Mm* versus those that were phagocytosed early at 3 dpi led us to suspect that phagocytosis influenced traversal across the BRB and progression to granuloma formation. To further dissect the kinetics of infection at the BRB (retinal vascular endothelium), we infected double-transgenic (cross-bred for *kdrl* and *mpeg1: BB*) larvae with wild-type (WT) *Mm*. We noted that until 3 dpi, 83.3% (10 of 12) of the *Mm* were extracellular and within the lumen of the blood vessel (Figure 3A,B). We also found *Mm* in the wall of the vessel at 3 dpi (Figure 3B). Only two *Mm* (both of which were extracellular) were also found to have already crossed over to the retinal tissue (non-vascular region). The first evidence of phagocytosis was noted at 4 dpi only after entering the retinal tissue, and only solitary macrophages with engulfed *Mm* were seen until 5 dpi (Figure 3C). Aggregation of macrophages into nascent granulomas was seen only at 6 dpi (Figure 3D).

### 2.2. Depletion of Circulating Monocytes Increases Rate of Ocular Infection

Intracellular organisms such as mycobacteria or cryptococci are known to rely primarily on professional phagocytic cells such as macrophages and dendritic cells for traversal of vascular endothelial barriers such as the BBB or the BRB [12,14,18]. This is known as the Trojan Horse mechanism and has been elegantly described in zebrafish models of central nervous system infections such as TB and cryptococcus [12,14]. We therefore hypothesized that depletion of the circulating macrophages with liposomal clodronate would reduce the rate of ocular infection, especially beyond 3 dpi, when BRB traversal typically occurs. To confirm the efficacy of the drug, we found that the number of circulating macrophages in treated larvae was significantly lower on all days until 6 dpi, compared to larvae treated with control liposomes (Figure 4A–C). Surprisingly, the rate of ocular infection in clodronate-treated larvae remained consistently higher than in the controls, even beyond 3 dpi, when the *Mm* crosses the BRB (Figure 4D). The mean percentage of larvae with eye infection in the clodronate-treated group (49.98 ± 9.57) was significantly higher than in the control group (38.73 ± 8.46) after 6 dpi (*p* < 0.03, Student’s *t*-test). However, the rate of granuloma formation amongst infected larvae was marginally lower in the clodronate-treated group (8%, two of 25 larvae) compared to the control group (14.3%, four of 28 larvae). We expect that the lack of contribution from circulating macrophages could be responsible for the lower rate of granuloma formation.

### 2.3. ESX-1 Secretion System Is Required for BRB Traversal and Granuloma Formation

Next, we sought to determine the influence of bacterial virulence factors on BRB traversal and subsequent granuloma formation. Here, we used RD1 mutant *Mm* [8,19], which lacks the ESX-1 gene locus-encoded secretory system required for secretion of several proteins, including early secreted antigenic target-6 (ESAT-6). Previous studies have demonstrated the role of ESX-1 in the invasion of brain endothelial cells during macrophage-independent crossing of the blood–brain barrier [12,19]. As expected, the frequency of ocular infection by RD1 mutant *Mm* remained low, ranging between 16.6% and 9.1% over a follow-up period of 10 dpi (Figure 5A). Larvae infected with mutant *Mm* survived longer (up to 15 dpi), likely due to the lower virulence of the infecting organism. We also repeated the macrophage depletion experiment with RD1 mutant *Mm*. Here, we noted that in contrast to WT *Mm*, the rate of infection was higher in the control group compared to the macrophage-depleted group, from 1 to 6 dpi (Figure 5B). This observation suggests that macrophages are vital for BRB traversal by less-virulent *Mm*.

Finally, we investigated whether absence of the ESX-1 secretion system influenced the kinetics of BRB traversal and granuloma formation in double-transgenic larvae. We noted a delay among mutant *Mm* in crossing the BRB (4–5 dpi) as well as in granuloma formation (7 dpi). In fact, only one of the five ocular infections showed some signs of aggregation, though not necessarily representative of a true granuloma. The macrophages within the granuloma appeared to be loosely arranged, unlike the compact arrangement noted with WT *Mm* (Figure 6A–D). To further establish this differential transit across the BRB, we injected an equally mixed population of WT and RD1 mutant *Mm* into *kdrl* larvae, where we again found that at 4 dpi, WT *Mm* had already crossed the BRB in 52.9% (9/17) of the ocular infections, while RD1 mutant *Mm* remained present in the blood vessels in all larvae with ocular infections (*n* = 4) (Figure 7).

Collectively, our data suggest an apparent dichotomy in the predominant mechanism of BRB traversal by virulent and avirulent bacteria. Virulent (RD1-competent) *Mm* appear to cross the BRB as free, extracellular bacteria using ESX-1-dependent mechanisms and progress to the formation of compact, early granulomas. Avirulent or less-virulent *Mm* require prior phagocytosis by circulating macrophages and cross the BRB using the Trojan Horse mechanism. These infections mostly do not progress to granuloma formation, and even if they do, typically result in loosely arranged granulomas.

## 3. Discussion

CNS invasion is generally caused by intracellular pathogens. Since the Trojan Horse hypothesis adequately explains their ability to cross CNS barriers, the extracellular pathways of these pathogens have mostly been ignored despite the fact that the same pathogens, while free or extracellular, can cross the CNS barriers nearly as, or sometimes even more, efficiently than while inside phagocytes [12]. In this study, while examining the interactions between mycobacteria and the BRB in a zebrafish model of ocular TB, we demonstrated that virulent (RD1-competent) mycobacteria typically adopt extracellular pathways and progress to granuloma formation in the eye. Avirulent bacteria require the Trojan Horse to get past the BRB and generally do not progress to granuloma formation.

The role of extracellular *Mtb* in CNS or retinal invasion is supported by several factors. Recent in vivo, in vitro, and transcriptional studies suggest the possibility of extracellular dissemination of *Mtb* from the lungs [20,21,22]. *Mtb* is also able to survive and replicate ex vivo in blood, where it develops enhanced virulence [23]. In vivo and in vitro studies have demonstrated that free *Mtb* invades vascular endothelial cells aided by secretion of virulence factors, such as ESAT-6 [12,19]. *Mtb* is also able to form a replicative niche within lymphatic endothelial cells [24]. Finally, the likelihood of extracellular dissemination of *Mtb* within the host can also be explained from an evolutionary perspective. The life cycle of *Mtb* depends on causing pulmonary disease and dissemination to a new host [25]. Therefore, it is unlikely that *Mtb* will adopt survival strategies within the phagocyte for the purpose of extrapulmonary dissemination.

Our data supporting hematogenous dissemination by extracellular *Mtb* are consistent with earlier studies on the zebrafish embryo model. For example, in the TB meningitis model, *Mm* were found in the brain tissue in all embryos, when phagocytes were similarly depleted [12]. In contrast, phagocyte depletion with the *pu.1* morpholino resulted in decreased ”extravascular” migration of *Mm* in another study [26]. However, in that study, the BRB/BBB that possess tight junctions, were not specifically identified, and therefore, the results are not truly divergent from our study. Our data are also not in conflict with the possibility of infected macrophages from the primary granuloma, facilitating its expansion by recruiting new macrophages and also disseminating into deeper tissues [27] or even to distal locations [28]. Neither process necessarily involves migration across the BRB/BBB, which is the primary context of our study. Other than BRB migration, we found that ESX-1 also influences the likelihood of granuloma formation within the retina. Here again, our results conform with earlier reports of virulent mycobacteria using ESX-1 for recruitment of new macrophages to the nascent granuloma, as well as facilitating their infection and mobility within the granuloma, thus facilitating their expansion [27].

Our results have implications for the early events that decide the clinical phenotype and time to onset of disease in ocular TB. Ocular TB has distinct phenotypes that are known to affect tissues either inside or outside the BRB, and occasionally both [29]. The former include retinitis and retinal vasculitis, whereas choroiditis, cyclitis, and iritis are examples of the latter. Based on our results, we expect that virulent, extracellular bacilli would cross the BRB more often while less-virulent or less-invasive bacterial phenotypes would either remain outside the BRB or cross it packaged inside macrophages. Alternatively, relatively avirulent phenotypes in a given host may enter non-professional phagocytes such as the retinal pigment epithelium (RPE), where they may remain “silent” for a prolonged period. The possibility of “silent” infection, at least in certain phenotypes of ocular TB, is supported by recent in vitro studies in RPE cells. Bacterial virulence genes such as ESX-1 are down-regulated within *Mtb*- infected RPE cells [30]. These cells limit intracellular *Mtb* replication and have prolonged survival even in the presence of infection [31]. We speculate that virulent transformation of such avirulent phenotypes over time could lead to initiation of the inflammatory cascade in such eyes.

The significance of the results of our study is limited by two critical missing links. First, we could not colocalise phagocytosed (intracellular) *Mm* within GFP-expressing endothelial cells during the stage of BRB invasion, thus failing to observe the actual event of crossing by extracellular *Mm*. The relationship between *Mm* and retinal vasculature could have been better delineated by orthogonal projections, but the resolution we obtained for these images was poor. Second, we were unable to demonstrate fluorescent-dye (FITC-dextran/sodium fluorescein) leakage at points of contact by *Mm* with the vascular endothelium. The larval coats were so thick at 3 dpi (7 dpf) that they did not allow entry of the injecting needle into the caudal vein. It is possible that some of the *Mm* would have been ingested by other cell types, specifically neutrophils. However, neutrophils do not interact with *Mm* at initial infection sites and only arrive later at the site of infection [32]. Besides, macrophage depletion by liposomal clodronate does not have any effect on neutrophil numbers [33]. Taken together, these points suggest it is unlikely that neutrophils played any role in BRB traversal by *Mm* in macrophage-depleted larvae.

Among other limitations, the ocular infection at 1 dpi (60%) could have been exaggerated due to the increased retinal vasculature at 4 dpf—the time point of larval infection in our study. Finally, the larval mortality rate at 6 dpi was significant at nearly 50% of the initial cohort (Figure 1A). This could be accounted for by the stress induced by the daily anesthesia and fluorescent microscopy evaluation and by the pathogenicity of *Mm* in these larvae. Notably, the non-pathogenic RD1 mutant-infected larvae survived up to 15 dpi in our experiments.

Nevertheless, our study provides reasonable evidence of the likelihood of dichotomous pathways with distinct host responses during mycobacterial invasion of the BRB. Our observations should also provide direction to the investigation of CNS barriers in meningeal TB as well as in other infections of the CNS.

## 4. Methods

### 4.1. Zebrafish Husbandry

Wild-type AB and various transgenic (Tg) zebrafish (*Tg (kdrl::EGFP,* green fluorescent endothelial cells), *Tg (mpeg1::BB,* red fluorescent macrophages), and double *Tg (KDRL::mpeg1,* red fluorescent macrophages with green fluorescent endothelial cells)) were maintained as previously described [8,32]. Eggs were obtained by natural spawning, and fertilized embryos were treated with 0.2 mM 1-phenyl-2-thiourea (PTU) (Sigma-Aldrich, P7629) to prevent melanophore formation. The guidelines recommended by the Committee for the Purpose of Control and Supervision of Experiments on Animals (CPCSEA), Government of India, were followed for zebrafish maintenance and experimentation.

### 4.2. Bacterial Strains

Wild-type *Mm* (WT/pmsp12: wasabi), WT/pmsp12::tdTomato (pTEC27 deposited with Adgene), WT/pmsp12::EBFP2, expressing green, red, and blue fluorescent proteins, respectively, and RD1 mutant described by Volkman et al. [8,32], were grown at 32 °C in Middlebrook’s 7H9 broth (M198, Himedia, Mumbai, India) supplemented with Middlebrook oleic acid/albumin/dextrose/catalase (OADC) and antibiotic hygromycin B (final concentration 50 mg/mL). Cultures and preparations of single cell *Mm* for infections were made as described [8] and were stored at a final dilution of 25 colony forming units (CFU)/nL in 5 μL and 10 μL aliquots, respectively, at −80 °C.

### 4.3. Caudal Vein Injections

Larvae were dechorionated at 3 dpf, and external feeding was started at 4 dpf. For caudal vein injections, dechorionated larvae were anesthetized with 0.02% (w/v) ethyl 3-aminobenzoate methanesulfonate (MS-222) (Tricaine, Sigma-Aldrich, A5040) supplemented with 0.2 mM PTU. A single aliquot of bacterial suspension was thawed and mixed with 0.25 µL of 20% phenol red as described [8]. For every new aliquot, the bacterial count per nL was checked under fluorescent microscopy, and the colony count checked by plating on Middlebrook’s 7H10 medium. It was also ensured that the same size of drop was used in all injections without changing the injector settings. At 4 days-post-fertilization (dpf), larvae were infected via caudal vein by injecting 1–2 nL single cell suspension containing ~25 pre-processed *Mm*. Bacterial injections were controlled using an Olympus SZ stereo-microscope with a Femtojet micro-injector (Eppendorf). Injected larvae were recovered by transferring them into fresh embryo media supplemented with 0.2 mM PTU. Physically damaged/non-viable embryos were discarded prior to injections. During the course of follow-up, larvae were transferred into fresh embryo media containing PTU every 24 h.

### 4.4. Microscopy

For live imaging, larvae were anesthetized with 0.02% (*w*/*v*) tricaine and screened for eye infection using an Olympus BX53 upright fluorescent microscope with z-stack attachment and Olympus Cellsense imaging software. For the infection analysis, larvae were anesthetized and mounted in 3% methyl cellulose and manipulated to place them laterally on the flat side to view one eye at a time [15]. Imaging was done with a 20× (U PlanFL N, 20×/0.50) objective along with FITC/TRITC/DAPI filters to detect green/red/blue fluorescent light, respectively. The presence of bacteria and infected eyes were scored manually, and early infection and granuloma were confirmed and analyzed visually. The primary outcome measures were frequency of ocular infection, timing of phagocytosis, and aggregation of macrophages into a granuloma.

### 4.5. Depletion of Circulating Monocytes

To deplete circulating monocytes selectively, 2–3 nL clodronate liposomes and phosphate buffered saline (PBS)-loaded control liposomes with 0.1% phenol red suspension were injected intravenously into the caudal vein of 3 dpf *Tg* (*mpeg1*: *BB*) larvae [33]. To prevent overwhelming infection and early death of the embryos, we did not use any anti-sense oligonucleotide (morpholino) directed against the myeloid transcription factor gene pu.1. Additionally, the persistence of tissue-resident macrophages in the retina allowed us to visualize the post-BRB migration events up to 6 dpi. At 1 day-post-injection (dpi), monocyte depletion in clodronate liposomes injected larvae was confirmed by loss of red fluorescence within the filter range (Cellsense mean fluorescence intensity [MFI]) and by counting manually under fluorescent microscopy [34].

## Figures and Tables

**Figure 1 pathogens-10-00333-f001:**
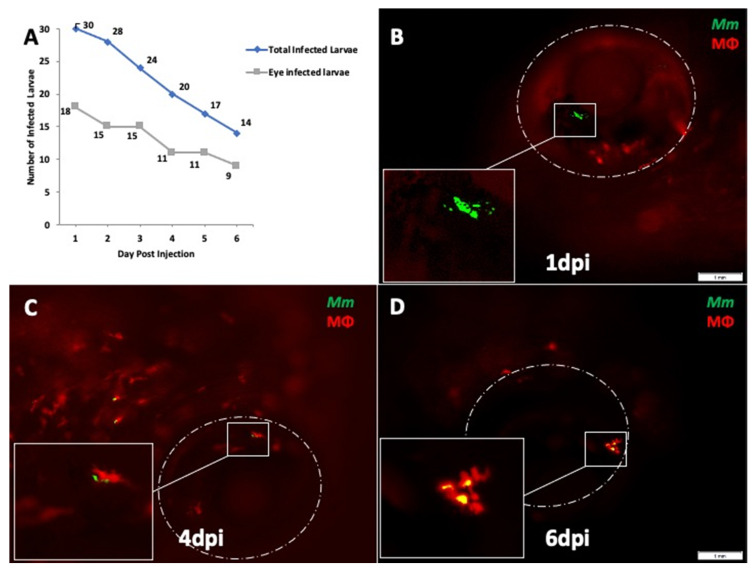
Ocular infection and granuloma formation with wild-type *Mycobacterium marinum* (*Mm*) in *mpeg1:BB* transgenic larvae. (**A**) Frequency of ocular infection following caudal vein infection with 25 colony-forming units (CFU) of *Mm*, 1–6 days post-injection (dpi). Granuloma formation was seen in six (33.3%) of the 18 eye-infected larvae. The kinetics of granuloma formation in all six eyes were as follows: (**B**) extracellular *Mm* (green) seeded in the eye at 1 dpi. The *Mm* remained extracellular until 3 dpi. (**C**) *Mm* phagocytosed inside a solitary macrophage (red) at 4 dpi. (**D**) Compact granuloma formation at 6 dpi, comprising macrophages with or without phagocytosed *Mm*. All images in the figure are from the same larva.

**Figure 2 pathogens-10-00333-f002:**
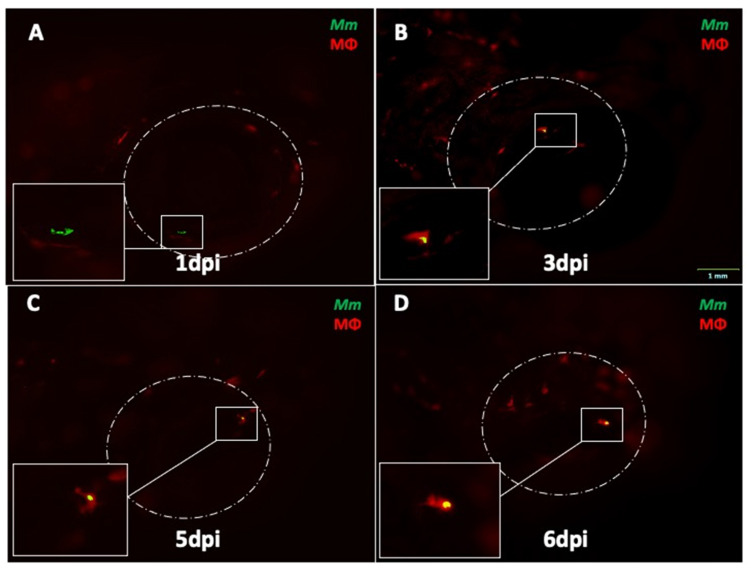
Progression of ocular infection in eyes with early phagocytosis in *mpeg1:BB* transgenic larvae. (**A**) Extracellular *Mycobacterium marinum* (*Mm*, green) in the eye at 1 dpi. (**B**) Phagocytosis (red macrophages) is seen early at 3 dpi. (**C**,**D**) *Mm* remain inside solitary macrophages until 6 dpi, with no evidence of aggregation into granuloma. This sequence of events was noted in four out of 18 eye-infected larvae. All images in the figure are from the same larva.

**Figure 3 pathogens-10-00333-f003:**
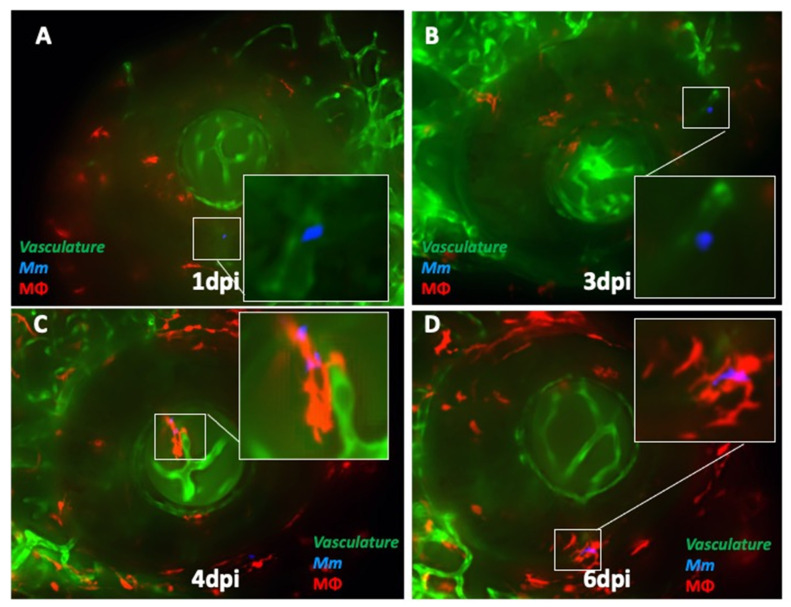
Kinetics of ocular infection and granuloma formation in double-transgenic (cross-bred for *kdrl* and *mpeg1: BB*) larvae with wild-type *Mycobacterium marinum* (*Mm)*. (**A**) Extracellular *Mm* (blue) lying in the lumen of retinal blood vessels at 1 dpi, and (**B**) crossing the vascular endothelium at 3 dpi. (**C**) The first appearance of phagocytosis inside the retinal tissue at 4 dpi, with the infected macrophage closely abutting the vascular endothelium, and (**D**) aggregation of macrophages into a granuloma within the retinal tissue at 6 dpi.

**Figure 4 pathogens-10-00333-f004:**
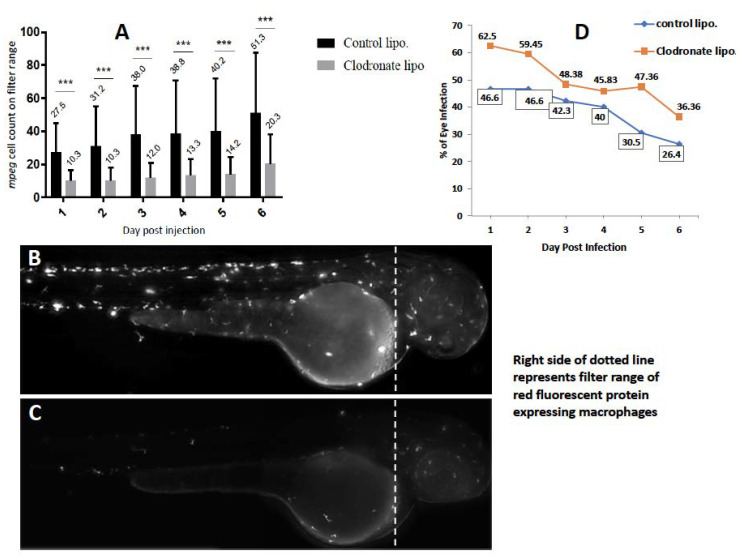
Effect of depletion of circulating monocytes on ocular infection. Liposomal clodronate was injected into *mpeg1:BB* transgenic larvae at 3 days post-fertilization (dpf), followed by wild-type *Mycobacterium marinum* injection at 4 dpf. Monocyte depletion within the head portion (shown by dashed line in **B**,**C**) was confirmed by loss of red fluorescence within the filter range at 4 dpf (Cellsense mean fluorescence intensity (MFI)) and additionally by counting manually under fluorescent microscopy. (**A**) MFI within the selected head area was significantly lower in the clodronate liposome-treated larvae, compared to the controls on days 1–6 post-injection (dpi) (*** *p* < 0.0001). (**B**) Representative fluorescent image (2x magnification) of control, and (**C**) clodronate liposome-treated larvae. (**D**) Higher ocular infection rate in clodronate liposome-treated larvae compared to controls on all days from 1–6 dpi. The mean percentage of larvae with eye infection in the clodronate-treated group (49.98 ± 9.57) was significantly higher than in the control group (38.73 ± 8.46) beyond 6 dpi (*p* < 0.03, Student’s *t*-test).

**Figure 5 pathogens-10-00333-f005:**
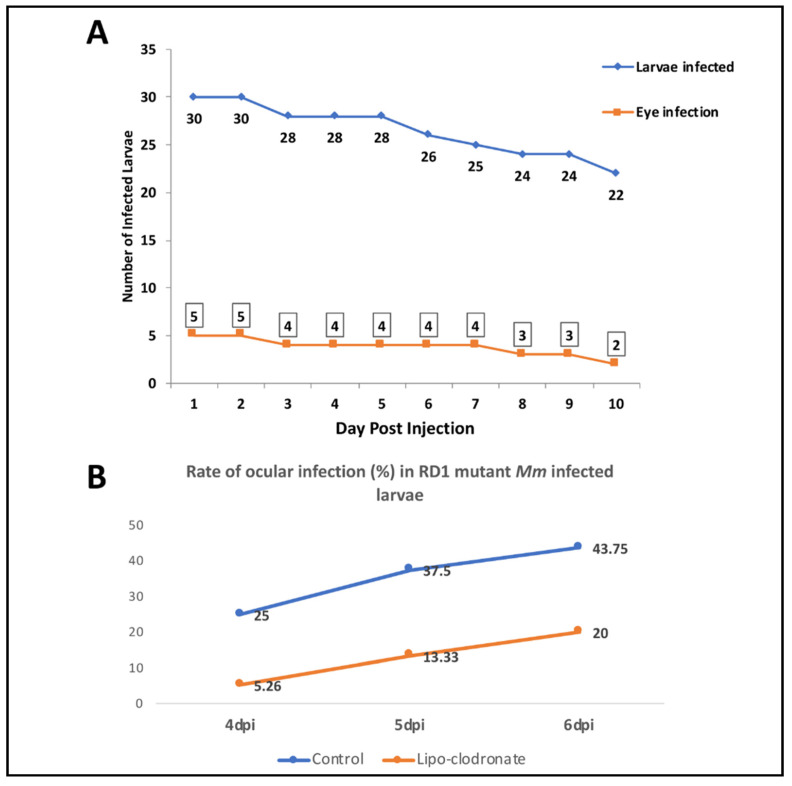
Effect of RD1 mutation in *Mycobacterium marinum* (*Mm*) on frequency of ocular infection. We infected four dpf larvae with 25 colony forming units (CFU) of RD1-mutant Mm. (**A**) Line graph showing significantly lower frequency of ocular infection with RD1 mutant *Mm* (compared with Figure 1A), even up to 10 dpi. The infected larvae, however, survived longer than during infection with wild-type (WT) *Mm*, as shown by the low slope of the graph. (**B**) Depletion of circulating monocytes with clodronate liposomes led to a drop in the ocular infection rate, the effect being opposite to that from infection with WT *Mm*.

**Figure 6 pathogens-10-00333-f006:**
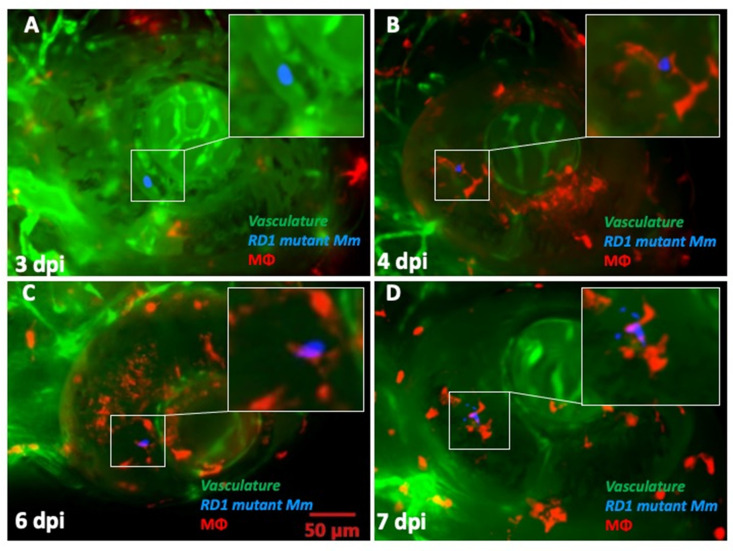
Kinetics of granuloma formation following ocular infection with RD1 mutant *Mycobacterium marinum* (*Mm*). (**A**) Extracellular *Mm* (blue) seen within retinal vascular lumen at 2 dpi. (**B**) Phagocytosed *Mm* (red macrophage) within the retinal tissue at 4 dpi. (**C**) Persistence of solitary macrophage with no evidence of aggregation even at 6 dpi. (**D**) Loose macrophage aggregation at 7 dpi, not representative of a true granuloma. This was seen in only one of the five ocular infections with RD1 mutant *Mm*.

**Figure 7 pathogens-10-00333-f007:**
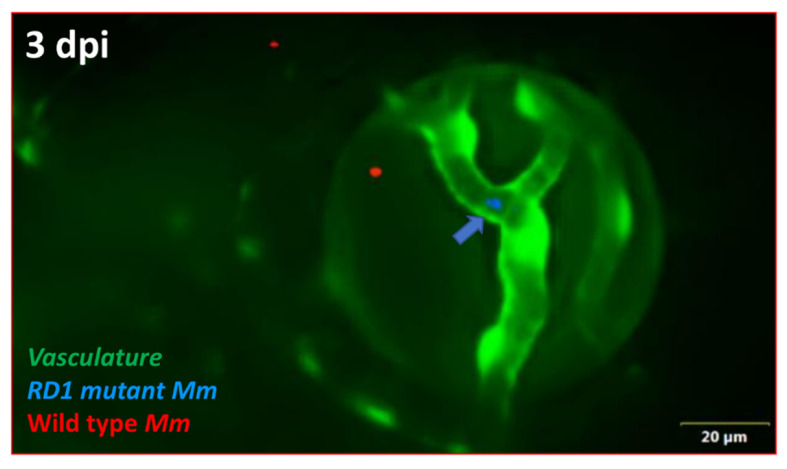
Differential progression of ocular infection by wild-type (WT) and RD1 mutant *Mycobacterium marinum* (*Mm*). At 3 dpi, WT *Mm* (red) have already crossed the vascular endothelial barrier and reached the retinal tissue, while RD1 mutant *Mm* (blue, arrow) remain inside the vascular lumen (*n* = 4/17).

## Data Availability

Data is contained within the article.

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
