# Peer review of "Role of Extracellular Mycobacteria in Blood-Retinal Barrier Invasion in a Zebrafish Model of Ocular TB"

_pathogens, 2021, doi:10.3390/pathogens10030333_

Round 1

Reviewer 1 Report

In this article, researchers employed a zebrafish model of ocular TB. According to their results, Low Mm inoculum was able to induce ocular infection, which was increased upon circulating monocytes depletion. Additionally, RD1 mutant Mm experiments suggested that WT Mm does not dependent on phagocytosis to neither cross BRB nor leading compact granuloma.

In Results 2.1, I would suggest starting with a brief introduction containing background information and experimental setting before mentioning experimental results. Otherwise, it could be difficult for readers to follow. In relation to results plot in figure1A, I have some questions:

  • How do you explain an infection rate of 60% at 1 dpi when used 25 CFU inoculum, while previous studies that used 100 CFU inoculum found only an infection rate of 20%? Could it be due to increased retinal vasculature at 4 dpf? Why did you consider infection at 4dpf instead 3dpf, as previous studies?
  • Is it normal to get a 50% larvae death rate at 6 dpi with 25 CFU inoculum? Might not be more physiological to reduce CFU inoculum?

I recommend rewriting paragraph corresponding to lines 75-94, it is difficult to follow.

In figure 5 the RD1 mutant Mm CFU employed in experiments it is not indicated.

In general, I would suggest increasing the low number of larvae and/or replicates. I am concern about the statistical significance of the presented results.

Author Response

Reviewer 1:

In this article, researchers employed a zebrafish model of ocular TB. According to their results, Low Mm inoculum was able to induce ocular infection, which was increased upon circulating monocytes depletion. Additionally, RD1 mutant Mm experiments suggested that WT Mm does not dependent on phagocytosis to neither cross BRB nor leading compact granuloma.

In Results 2.1, I would suggest starting with a brief introduction containing background information and experimental setting before mentioning experimental results. Otherwise, it could be difficult for readers to follow.

Re: We have added background information and details of experimental settings at the beginning of Result 2.1 that will help readers to grasp the experimental results.

In relation to results plot in figure1A, I have some questions:

  • How do you explain an infection rate of 60% at 1 dpi when used 25 CFU inoculum, while previous studies that used 100 CFU inoculum found only an infection rate of 20%? Could it be due to increased retinal vasculature at 4 dpf? Why did you consider infection at 4dpf instead 3dpf, as previous studies?

Re: We understand the reviewer’s concern on the increase in ocular infection rate at 1 dpi even after decreasing the dose of inoculum. Several factors could account for this paradox. One, as mentioned by the reviewer, could be the increased vascularity in the retina at 4 dpf as compared to 3 dpf. Another factor, that we noted in our previous study as well as in the current study (lines 95-97), is that bacteria could just be transiting through the retinal vasculature, at the time of observation, thereby erroneously increasing the infection count at 1 dpi. This possibility becomes more likely with the increased vasculature at 4 dpf. We have added this explanation in lines 266-8.

We considered inoculation at 4 dpf instead of 3 dpf since the focus of the current study was the mycobacterial invasion of blood-retinal barriers and we wanted additional surety on the anatomical development of these barriers. Besides, the TB meningitis model had also used inoculation at 4 dpf (van Leeuwen et al., Cell Microbiol., 2018)

  • Is it normal to get a 50% larvae death rate at 6 dpi with 25 CFU inoculum? Might not be more physiological to reduce CFU inoculum?

Re: The 50% larval death rate at 6 dpi is indeed high and could be influenced by several factors. The most important factor in our study could have been the stress induced by the daily anaesthesia and evaluation under the fluorescent microscope, in the infected fish. Besides, Mycobacterium marinum (Mm) are pathogenic to fish and even 25 CFU could be a potentially lethal inoculum for the larvae beyond a certain time period. Incidentally, the RD1 infected fish survived beyond 15 dpi in our experiments. In any case, we have added this point in the limitations section (lines 268-72).

We checked the literature on mortality rates in Zf larvae after Mm injection, but could not locate any objective data on the same. In the meningitis model the infected larvae had been followed till 5 dpi (9 dpf), as compared to 6 dpi in our study.

I recommend rewriting paragraph corresponding to lines 75-94, it is difficult to follow.

Re: We have rewritten lines 75-94 (presently lines 78-98) for easy understanding of the readers.

In figure 5 the RD1 mutant Mm CFU employed in experiments it is not indicated.

Re: We used 25 CFU even for the RD1 mutant Mm. This information has been added in the legend for Figure 5.

In general, I would suggest increasing the low number of larvae and/or replicates. I am concerned about the statistical significance of the presented results.

Re: We understand the reviewer’s concern on the apparently low number of larvae/replicates. However, as highlighted in the revised manuscript, we infected 30 larvae each in two separate experiments, that were followed up for rate of infection till 6 dpi. The numbers in subsequent experiments were limited by the rate of ocular infection followed by the rate of granuloma formation.

Reviewer 2 Report

-Line 60: “there was a steady decrease in viable embryos due to the Mm infection”

Considering the low infection dose (25 cfu/nl) used in this study I am concerned about the decrease in viable embryos. Knowing this infection model and knowing that other researchers in the field inoculate x10 higher Mm doses (even up to 300 cfu/larvae) with no effect on mortality, how it comes that the authors justify the mortality of the embryos due to the infection while using a low dose of Mm?

-Line 61: “injected the larvae at 4 dpf”

Are these larvae being feed from 5 dpf onwards? This information needs to be clearly stated in Methods as it is crucial for the interpretation of the results obtained. My concern is that this might be related to the mortality observed during the experiments. To avoid a starvation state, from 5 dpf onwards larvae would need external feeding due to the exhaustion of the yolk. This is crucial and could affect the results shown in this study as it is well known that starvation induces autophagy and autophagy plays a fundamental role during Mm infection. Reaching 10 dpf without external feeding would explain the mortality observed. The authors should provide a mortality curve of infected vs non-infected zebrafish from 4dpf to 10 dpf (=6dpi) to rule out this possibility.

-Line 80: “In the current study, granuloma formation was seen in 6 (33.3%) of the 18 eye-infected embryos”

How is a granuloma defined according to the authors? Is a group of 2 macrophages a granuloma? Terms like granuloma-like structure or early granuloma are also accepted in the field

-Line 82: “Mm remained extracellular”

How are the authors certain that Mm is extracellular? Could Mm not be inside a cell other than a macrophage?

-Line 86: “Finally, on 6 dpi, aggregation 86 of macrophages into granulomas was noted (figure 1D). The granulomas were typically compact with tight arrangement of macrophages, with or without Mm inside them”

            See my comment about line 80. 2 macrophages are visible in 1D, is this enough to state that this is a granuloma?

-Lines 103-109:

            These claims would definitely require to see an orthogonal projection of panels 3A and 3B. The position of Mm relative to the vasculature cannot be inferred by these max. projection images. Same applies for the phagocytosis claims in 3C.

-Lines 131-135: “However, the rate of granuloma formation amongst the infected larvae was marginally lower in the clodronate 132 treated group (8%, 2 of 25 larvae) as compared to the control group (14.3%, 4 of 28 larvae). We expect that the lack of contribution from circulating macrophages could be responsible 134 for the lower rate of granuloma formation”

            Is not what you would expect if you deplete macrophages and you are defining granulomas as group of infected macrophages? Should not be the conclusion here that the presence of macrophages reduces the infection most likely by killing Mm before the bacteria reaches the eye or that Mm could be reaching the eye inside a different type of cell (like neutrophils por example)?

-Lines 171-173: “The macrophages within the granuloma appeared to be loosely arranged unlike the compact arrangement noted with WT Mm (figure 6A-D)”

            This claim is not obvious from the images, maybe new panels with the “granulomas” side by side would help the reader. 3D projections would be awesome, this can be performed using Fiji.

-Figure 1:

            Are the images shown in B-D coming from the same larvae? This should be stated in the figure legend.

-Figure 2:

Are the images shown in A-D coming from the same larvae? This should be stated in the figure legend.

Could the authors add and indication of how many fish (X) out of the total fish (Y) present this phenotype? X out of Y or X/Y in the panels would be really helpful for the reader.

-Other concerns:

How many times these experiments have been repeated? Can the authors show how repetitive these results are through graphs displaying several repeats (mean+sem)?

Author Response

Reviewer 2:

-Line 60: “there was a steady decrease in viable embryos due to the Mm infection”

Considering the low infection dose (25 cfu/nl) used in this study I am concerned about the decrease in viable embryos. Knowing this infection model and knowing that other researchers in the field inoculate x10 higher Mm doses (even up to 300 cfu/larvae) with no effect on mortality, how it comes that the authors justify the mortality of the embryos due to the infection while using a low dose of Mm?

Re: We agree that the mortality rate of infected larvae was relatively high in our experiments. The most likely factor is the stress induced by repeated anaesthesia and examination under fluorescent microscope. We could not find exact data on larval mortality rates in publications on the TB meningitis model. In this model, the larvae were followed up till 5 dpi (9 dpf) as compare to 6 dpi in our study. However, it is possible that other researchers had lower mortality rates even with higher Mm inoculation due to differences in experimental protocols. In addition, Mm are pathogenic in fish and the infection would have influenced the mortality rate. Notably, the RD1-mutant infected larvae survived for up to 15 dpi in our experiments. In any case, we acknowledge that this is a significant limitation in our study, and have now included it in the limitations section our discussion (lines 268-72).

-Line 61: “injected the larvae at 4 dpf”

Are these larvae being fed from 5 dpf onwards? This information needs to be clearly stated in Methods as it is crucial for the interpretation of the results obtained. My concern is that this might be related to the mortality observed during the experiments. To avoid a starvation state, from 5 dpf onwards larvae would need external feeding due to the exhaustion of the yolk. This is crucial and could affect the results shown in this study as it is well known that starvation induces autophagy and autophagy plays a fundamental role during Mm infection. Reaching 10 dpf without external feeding would explain the mortality observed. The authors should provide a mortality curve of infected vs non-infected zebrafish from 4dpf to 10 dpf (=6dpi) to rule out this possibility.

Re: We agree with the reviewer’s suggestion that a starvation state post 5 dpf can induce increased mortality in larvae. Indeed, since we dechorionated the larvae at 3 dpf, the yolk sac would have been consumed faster than in normal growing larvae. To counter this starvation state, we provided external feeding to the larvae from 4 dpf onwards, with  commercially available water surface floating fine powder – PL-150 (PL: post larvae). We have added this point in line 297.

-Line 80: “In the current study, granuloma formation was seen in 6 (33.3%) of the 18 eye-infected embryos”

How is a granuloma defined according to the authors? Is a group of 2 macrophages a granuloma? Terms like granuloma-like structure or early granuloma are also accepted in the field

Re: We agree with the author’s concern that a group of two macrophages cannot be defined as granuloma. The term early granuloma (Davis and Ramakrishnan, Cell, 2009) is more appropriate to describe these aggregates. These aggregates have histological characteristics and gene expression that match adult granuloma (Davis et al., Immunity, 2002). In the revised manuscript, we have replaced the term granuloma with early granuloma. We have included the two-cell aggregate in Fig 1D in order to present sequential images from the same larva in the figure.

-Line 82: “Mm remained extracellular”

How are the authors certain that Mm is extracellular? Could Mm not be inside a cell other than a macrophage?

Re: It is possible that some of the Mm would have been ingested by other cell types, specifically neutrophils. However, neutrophils do not interact with Mm at initial infection sites and only arrive later at the site of infection (Yang et al., Cell Host Microbe, 2012). Besides, macrophage depletion by liposomal clodronate does not have any effect on neutrophil numbers (Bojarczuk et al., Sci Rep., 2016). Taken together it is unlikely that neutrophils had any role in crossing of BRB by Mm in macrophage depleted larvae. We thank the reviewer for raising this point and have added it to the discussion section (lines 260-5).

-Line 86: “Finally, on 6 dpi, aggregation of macrophages into granulomas was noted (figure 1D). The granulomas were typically compact with tight arrangement of macrophages, with or without Mm inside them”

            See my comment about line 80. 2 macrophages are visible in 1D, is this enough to state that this is a granuloma?

Re: As mentioned above, we have revised the manuscript using the term ‘early granuloma’ instead of granuloma.

-Lines 103-109:

            These claims would definitely require to see an orthogonal projection of panels 3A and 3B. The position of Mm relative to the vasculature cannot be inferred by these max. projection images. Same applies for the phagocytosis claims in 3C.

Re: We agree with the reviewer’s suggestion that orthogonal projections of panels 3A-C would have helped in better delineation of position of Mm in relation to retinal vasculature. However, the z-resolution of our images was not adequate for sufficient resolution of the orthogonal projection. We have acknowledged this limitation in the discussion section (lines 253-5).

-Lines 131-135: “However, the rate of granuloma formation amongst the infected larvae was marginally lower in the clodronate treated group (8%, 2 of 25 larvae) as compared to the control group (14.3%, 4 of 28 larvae). We expect that the lack of contribution from circulating macrophages could be responsible for the lower rate of granuloma formation”

            Is not what you would expect if you deplete macrophages and you are defining granulomas as group of infected macrophages? Should not be the conclusion here that the presence of macrophages reduces the infection most likely by killing Mm before the bacteria reaches the eye or that Mm could be reaching the eye inside a different type of cell (like neutrophils for example)?

Re: The reviewer has raised a valid point here. In our earlier study (Takaki et al., 2018), we had demonstrated that intraretinal granuloma are formed by resident macrophages (microglia) and circulating macrophages. Expectedly, depletion of the circulating macrophages decreased the rate of granuloma formation in the eye.

The other issue relates to eye infection in the absence of macrophages. We noted a significantly higher rate of eye infection after macrophage depletion. We have attributed these infections to extracellular Mm. It is quite likely that depletion of circulating macrophages would have increased the Mm population in the circulation thereby increasing their chances of infecting the eye. However, as discussed above, it is unlikely that other Mm could be residing inside other cell types, specifically neutrophils.

-Lines 171-173: “The macrophages within the granuloma appeared to be loosely arranged unlike the compact arrangement noted with WT Mm (figure 6A-D)”

            This claim is not obvious from the images, maybe new panels with the “granulomas” side by side would help the reader. 3D projections would be awesome, this can be performed using Fiji.

Re: We would have also liked to have 3D projections of the granuloma, but were limited by available equipment. We hope to attempt this in future experiments.

-Figure 1:

            Are the images shown in B-D coming from the same larvae? This should be stated in the figure legend.

Re: Yes, the images B-D are coming from the same larvae. This information has been added to the legend.

-Figure 2:

Are the images shown in A-D coming from the same larvae? This should be stated in the figure legend.

Could the authors add an indication of how many fish (X) out of the total fish (Y) present this phenotype? X out of Y or X/Y in the panels would be really helpful for the reader.

Re: The images in Fig 2 (A-D) are from the same larvae. This information is now added in the Fig legends. We have also added the X/Y information in the legends.

-Other concerns:

How many times these experiments have been repeated? Can the authors show how repetitive these results are through graphs displaying several repeats (mean+sem)?

Re: The experiment on ocular infection rate was repeated twice in groups of 30 larvae each. It was not possible to represent the result from two experiments in mean±sem.

Round 2

Reviewer 2 Report

Major concerns has been addressed and limitations have been added to the discussion.

This manuscript is a resubmission of an earlier submission. The following is a list of the peer review reports and author responses from that submission.